# Unraveling the Connections: Eating Issues, Microbiome, and Gastrointestinal Symptoms in Autism Spectrum Disorder

**DOI:** 10.3390/nu17030486

**Published:** 2025-01-29

**Authors:** Natalia Tomaszek, Agata Dominika Urbaniak, Daniel Bałdyga, Kamila Chwesiuk, Stefan Modzelewski, Napoleon Waszkiewicz

**Affiliations:** Department of Psychiatry, Medical University of Bialystok, pl. Wołodyjowskiego 2, 15-272 Białystok, Poland; 38778@student.umb.edu.pl (N.T.); 38780@student.umb.edu.pl (A.D.U.); daniel.baldyga6@gmail.com (D.B.); 37059@student.umb.edu.pl (K.C.); napoleon.waszkiewicz@umb.edu.pl (N.W.)

**Keywords:** autism spectrum disorder, avoidant/restrictive food intake disorder, microbiome–gut–brain axis, gastrointestinal symptoms, gut microbiota, dysbiosis, dietary preferences, food selectivity, dietary interventions, eating disorders

## Abstract

Autism spectrum disorder (ASD) is a neurodevelopmental condition characterized by challenges in social communication, restricted interests, and repetitive behaviors. It is also associated with a high prevalence of eating disorders, gastrointestinal (GI) symptoms, and alterations in gut microbiota composition. One of the most pressing concerns is food selectivity. Various eating disorders, such as food neophobia, avoidant/restrictive food intake disorder (ARFID), specific dietary patterns, and poor-quality diets, are commonly observed in this population, often leading to nutrient deficiencies. Additionally, gastrointestinal problems in children with ASD are linked to imbalances in gut microbiota and immune system dysregulation. The aim of this narrative review is to identify previous associations between the gut–brain axis and gastrointestinal problems in ASD. We discuss the impact of the “microbiome–gut–brain axis”, a bidirectional connection between gut microbiota and brain function, on the development and symptoms of ASD. In gastrointestinal problems associated with ASD, a ‘vicious cycle’ may play a significant role: ASD symptoms contribute to the prevalence of ARFID, which in turn leads to microbiota degradation, ultimately worsening ASD symptoms. Current data suggest a link between gastrointestinal problems in ASD and the microbiota, but the amount of evidence is limited. Further research is needed, targeting the correlation of a patient’s microbiota status, dietary habits, and disease course.

## 1. Introduction

Autism spectrum disorder (ASD) is a group of neurodevelopmental conditions characterized by difficulties in social interaction and communication, alongside restricted or repetitive behaviors [1]. The median prevalence of ASD is estimated at 0.72% [2], and recent studies report a continued rise over time, both at the national level and within specific subgroups [3]. While the core features of ASD are well-established, there is increasing recognition of the presence of feeding disorders, gastrointestinal issues, and changes in gut microbiota among affected individuals [4]. Common gastrointestinal symptoms include constipation, abdominal pain, and diarrhea [5]. Additionally, research indicates a significant disparity in feeding problems between children with and without ASD, with those in the former group being approximately five times more likely to experience such issues [6]. Moreover, patients with ASD often exhibit a high incidence of selective eating patterns, such as food neophobia and food selectivity, which are both components of avoidant/restrictive food intake disorder (ARFID) [7,8]. These eating difficulties are frequently influenced by behavioral rigidity (difficulty in adapting to changes in routine, environment, or expectations, leading to repetitive and restricted behaviors [9]), sensory sensitivities, and anxiety—key characteristics of ASD [10].

The complex interplay between feeding problems, gastrointestinal symptoms, and the gut–brain axis in children with ASD presents a critical area of investigation. By examining these intricate interactions, we aim to answer the question: what is the link between gastric symptoms, ASD, and microbiota?

## 2. Materials and Methods

The aim of this narrative review was to describe evidence for a three-way interaction between ASD, eating issues, and gut microbiota alterations, specifically examining how food selectivity patterns in ASD affect gut microbiota composition and its potential feedback effects on ASD symptoms through the gut–brain axis.

We included studies from the years 2000–2024. During the initial search, we included both original and review articles to identify potential records and original studies during the screening process. The PubMed database was searched on 31 November 2024, for studies published between 2004 and 2024 using the following search terms and their combinations: “ARFID, genetics, food selectivity, sensory sensitivity, food neophobia, ASD, children, mealtime, dietary patterns, diet quality, gastrointestinal symptoms, vitamin deficiency, micronutrients, metabolic syndrome, gut microbiota, microbiome, microbiota-gut-brain axis”.

Subsequently, duplicates were removed using EndNote 21 software (some studies appeared multiple times due to the use of various combinations of search terms). Since this work is a narrative review, we did not establish strict inclusion criteria for the articles. During the screening process, we included studies that focused on nutritional problems in children with ASD, as well as selected studies describing this issue in animal models. Based on titles and abstracts, the authors excluded studies that focused strictly on adults and conference abstracts. We also excluded studies not written in English. We also included supporting literature covering possible therapies.

Our flexible search strategy reflects the need to synthesize information from diverse sources due to the substantial heterogeneity of human studies, which precludes further synthesis. Consequently, we emphasize that this work is a narrative review, which constitutes its primary limitation.

## 3. Eating Disorders Associated ASD

Feeding abnormalities in individuals with ASD can begin early in life, even during breastfeeding. Common challenges during this period include disorganized and vigorous sucking that continues despite the child being satiated [11]. A study by Provost et al. found that children with ASD often struggle to eat in various settings, with over half experiencing difficulties dining at regular restaurants and facing challenges when eating at friends’ homes [12]. These issues can manifest in reactions such as screaming, crying, and tantrums when encountering unknown foods or when others are eating [13].

Another category of eating-related difficulties is food selectivity (FS), commonly defined by three domains: food refusal, limited food variety, and high-frequency single food intake [14]. Rejection can stem from food sensory factors or preferences for its appearance [15]. Molina-López et al. compared 55 ASD children to 91 neurotypical (NT) peers using the Food Frequency Questionnaire (FFQ) [16]. Results showed FS in 60.6% of the ASD group versus 37.9% of the NT group, with FS defined as rejecting over 33% of offered foods. Similarly, Sharp et al. found that 78% of 70 ASD children omitted one or more food groups, most often vegetables (67.1%) [15]. Bandini et al. also observed this trend in a study comparing 53 ASD and 58 NT children aged 3–11, where ASD children rejected 41.7% of foods compared to 18.9% in NT peers [17].

One of the key components of food selectivity is food neophobia (FN). FN is defined as a reluctance or even fear of trying new foods [18]. While FN is also common in NT children, it often is transient for them. ASD children, on the other hand, tend to present this behavior constantly over time [19]. Wallace et al. state that FN is more prevalent within the ASD population than in NT children [20]. Interestingly, the paper shows that FN in the NT population leads to lower Body Mass Index (BMI), whereas when combined with ASD, it leads to higher BMI. Kral et al. confirm that ASD children are significantly more reluctant to try unfamiliar food [21].

## 4. Avoidant/Restrictive Food Intake Disorder (ARFID)

Avoidant/restrictive food intake disorder is a clinically diagnosable type of food selectivity, characterized by significant limitations in both the quantity and variety of food consumed. However, unlike FS or FN, to be diagnosed, it must result in malnutrition, substantial weight loss, or nutritional deficiencies, often requiring reliance on nutritional supplements or tube feeding to maintain health. Unlike anorexia nervosa (AN) or bulimia nervosa (BN), ARFID is not associated with body image concerns or a desire to be thin. In addition to physical health consequences, the disorder can cause notable difficulties in personal, social, educational, or occupational functioning, particularly when social eating situations provoke distress or avoidance [22].

The prevalence of ARFID in the general population is not well established, but estimates suggest it may affect between 0.5% and 5% of people [23]. Research has shown that, in comparison to individuals with AN or BN, patients with ARFID tend to be younger, are more likely to be male, and are often diagnosed with co-occurring psychiatric and/or medical issues [24].

In a comprehensive, nationwide study conducted in Sweden involving 16,951 twin pairs aged 6 to 12 years, findings revealed a high heritability for ARFID, indicating that genetic factors significantly contribute to the disorder’s development. Additionally, the study found that non-shared environmental factors played a smaller yet significant role [25]. Koomar et al. conducted a genome-wide association study to identify genetic variants associated with ARFID. The study found one genome-wide significant single nucleotide polymorphism on chromosome 5 near the gene ZSWIM6. This gene has been previously implicated in neurodevelopmental conditions like cognitive impairment and schizophrenia [26].

ARFID often presents with characteristics similar to those observed in children and adolescents with ASD. Among children diagnosed with ARFID, the prevalence of ASD ranges from 8.2% to 54.8% [27]. In contrast, the prevalence of ARFID in the general pediatric population is significantly lower, estimated at approximately 3.2% based on self-reported data from primary school children [28].

### 4.1. ARFID Subtypes

DSM-5 [29] and ICD-11 [30] identify three main subtypes of ARFID: lack of interest, sensory–sensitivity-based and fear/aversive. A fourth subtype, combined presentation, may be diagnosed when multiple causal factors are present.

The lack of interest subtype often experiences difficulties with the act of eating, such as taking small bites and requiring prolonged time to finish meals. The sensory–sensitivity subtype exhibits a restricted variety of foods due to sensory issues and aversions to certain items. This group usually displays profound rigidity in their eating behaviors, manifesting as food selectivity and food neophobia. The aversive subtype develops avoidance behaviors due to specific experiences or fears, such as choking or vomiting. This pattern of nutritional restriction often evolves from a traumatic event or a deep-seated fear of choking, pain, or nausea, leading to significant anxiety surrounding food intake. However, this subtype may represent a distinct and mutually exclusive category, showing no overlap with the characteristics of other presentations [31].

Sanchez-Cerezo et al. performed a surveillance study utilizing latent class analysis (LCA) [32]. They assessed the following six binary factors: lack of appetite, lack of interest in eating or food, difficulties with practicalities of feeding behaviors (e.g., small bites or slow eating), sensory sensitivity (e.g., sensitivity to taste, smell, color, or texture), rigid eating behaviors (e.g., brand-specific preferences or food items cannot touch on the plate), and fear of aversive consequences of eating (e.g., choking or vomiting). Researchers classified 319 children and adolescents in the UK and Ireland into the four mentioned subtypes of ARFID. The combined subtype was the most prevalent (38.2%), characterized by high probabilities of all assessed symptoms except fear of eating. The Sensory subtype (29.5%) displayed predominantly sensory sensitivity and rigid eating behaviors, while the Lack of Interest subtype (25.1%) manifested through a pronounced lack of appetite and food interest as well as medium practical difficulties. The fear subtype, the least common one (7.2%), presented with strong fear of aversive consequences, shorter symptom duration, and elevated anxiety levels. It is worth noting that analysis of the classes revealed that the combined and sensory subtypes had significantly more ASD occurrences than the other classes.

According to Eddy et al., the distribution is slightly different—the predominant subtype of ARFID is characterized by insufficient food intake and a lack of interest in feeding, accounting for 57.6% of cases [33]. The second most common subtype involves a limited diet based on sensory preferences, comprising 21.2% of cases. Conversely, the subtype associated with aversive or traumatic experiences is relatively rare.

The most common subtype of ARFID in individuals with ASD is linked to heightened sensory sensitivities [34]. Although patients with ASD display increased sensitivity to sensory aspects of food and show reduced interest in eating, they do not significantly differ from non-ASD patients in their concern about aversive consequences, such as fear of choking or vomiting [35] (Table 1).

### 4.2. ARFID Impact on Gut Microbiome

ARFID affects the gut microbiota by altering microbial diversity and composition. Children with ARFID exhibit changes in gut microbiota diversity, showing higher diversity indices compared to healthy controls. There is an increased abundance of potentially pathogenic bacteria such as Enterobacterales and Bacteroidaceae—including species like *Bacteroides vulgatus*—and a significant reduction in beneficial *Bifidobacterium* species. This dysbiotic state suggests an imbalance between harmful and beneficial bacteria, potentially affecting the gut–brain axis (further expanded below in Section 9 and Section 10) [36].

## 5. Impact of Sensory Processing Issues, Behavioral Rigidity, and Anxiety on Food Preferences

A key characteristic of autism is the presence of Restrictive and Repetitive Behaviors (RRB) [37]. In addition, autistic children often exhibit sensory processing issues, such as atypical sensory modulation (ASM), where they respond disproportionately to sensory stimuli [38]. Patients with ASD typically exhibit much higher levels of oral sensory over-sensitivity (SOR) compared to those without ASD [39].

Because of that, children with ASD may have strong reactions to specific food attributes such as smell, taste, appearance, temperature, texture, or composition. These sensory oversensitivities could be one of the reasons why FS is so common among ASD children [19]. A study by Byrska et al. compared the FS patterns of children with ASD to NT children based on various food traits [40]. The authors found that ASD children are more likely to refuse food that is sour, sticky, or when ingredients are mixed or touched on the plate, compared to NT children. Similarly, Hubbard et al. report that ASD children often reject food based on texture or consistency and when ingredients are combined, while these factors are less significant for NT children [41].

Intolerable food consumption often leads to refusal behaviors such as crying, pushing away the feeder or spoon, turning the head, blocking the mouth, and, in some instances, more severe reactions like fleeing from the table, aggression, or self-harm [42].

A longitudinal study by Suarez et al. [43] and a follow-up [44] examined sensory over-responsivity (SOR) and FS over time. Using a custom SOR questionnaire and the Repetitive Behaviors Scale–Revised (RBS-R) for repetitive behaviors (RRB), they found that SOR was significantly linked to FS, with this relationship persisting over two years. Increased SOR also predicted more severe FS, but RRB was not an independent predictor, suggesting a strong correlation between SOR and RRB. Bandini et al. also confirm that food repertoire in ASD patients does not increase over time [45].

Eating disorders, particularly food selectivity, food neophobia, and avoidant/restrictive food intake disorder, are highly prevalent among individuals with autism spectrum disorder due to sensory processing issues, behavioral rigidity, and anxiety. These challenges often result in a limited dietary variety, persistent nutritional deficiencies, and social difficulties, with sensory sensitivities playing a critical role.

## 6. Dietary Patterns in Children with ASD

It is well documented that a properly varied diet greatly impacts children’s cognitive and physical development [46]. Unfortunately, parents of children with ASD regularly face significant challenges in providing a balanced and nutritious diet for their children [8].

The initial stages of feeding are significant for a child’s growth and development and require considerable attention in the context of ASD [47]. Xiang et al. conducted a comprehensive investigation into the feeding patterns of children with ASD and their typically developing (TD) counterparts. The researchers observed that children diagnosed with ASD were breastfed for a shorter duration than TD controls, with an average of 8 months compared to 10 months, respectively. An analysis of results from the Autism Behavior Checklist (ABC) and the Childhood Autism Rating Scale (CARS) revealed that children who were breastfed for 12 months or more exhibited lower scores than those who were breastfed for less than 6 months. Furthermore, ASD-diagnosed children were introduced to complementary foods at a later stage and demonstrated a lower acceptance rate. This resulted in longer periods of bottle-feeding [48].

A questionnaire study conducted by Emond et al. indicated that individuals with ASD exhibited feeding difficulties from infancy and had diets that were less diversified from the age of 15 months. At six months of age, parents reported challenges with the introduction of solid foods and described their children as “slow eaters”. Additionally, children consumed less fruit and vegetables, sweets, and carbonated beverages. However, no differences were observed in weight, height, or BMI at 18 months and seven years between the ASD patients and the control group [49].

The findings of a study by Evans et al. also indicated a lower consumption of fruits and vegetables in the ASD group compared to typically developing children. Contrary to findings in the above-mentioned study, autistic children displayed a preference for snack foods and dairy-free sweetened soft drinks [50].

The research group of Plaza-Diaz et al. identified a comparable pattern in their investigation. A lower intake of fruits and vegetables was reported in the ASD children. Foods of high fat and high energy content, including juices, snack foods, confectionery, and baked goods, characterized the subjects’ dietary choices. Additionally, an increase in dairy consumption was associated with a greater intake of cereals and pasta. Children exhibited lower consumption of lean meat and eggs, with excessive intake of fatty meats. Parents also reported that their children were more likely to eat meat when it was shredded [51].

A further study conducted by Raspini et al. indicated that preschoolers diagnosed with ASD consumed a greater quantity of soft drinks, sweetened fruit juices, and snacks (including cookies, puddings, crackers, breadsticks, fried fish, and French fries) than age-matched peers with TD. The participants consumed greater quantities of yogurt, milk, and red meat. Conversely, their intake of raw vegetables, fruits, and cereals diminished, reducing fiber intake. Children with ASD also exhibited increased consumption of simple carbohydrates and processed foods [52].

In their study, Malhi et al. observed that children with ASD demonstrated a reduced intake of a range of food items, including fruits and vegetables (such as spinach, tomatoes, potatoes, oranges, and papaya) and proteins [53].

The study by Diolordi et al. identified a statistically significant reduction in milk and yogurt consumption among children with ASD compared to the control group. Additionally, researchers noted a marked decrease in vegetable intake within the ASD group, with 41% of children (mean age 58.3 months) never consuming vegetables, in contrast to only 6.9% of TD children. A significantly higher frequency of rice and pulse consumption was also observed among children with ASD. Additionally, the individuals ingested greater quantities of snacks, fruit juices, pasta, pizza, and biscuits. There was no notable divergence in fruit intake between the two cohorts. Although children with ASD consumed less meat and fish overall, these differences were not statistically significant. Notably, 27.3% of children with ASD had never tried fish, compared to none in the TD group. Such aversions may be attributed to the flavor of fish or the presence of bones [54].

Schreck et al. conducted an in-depth investigation into the specific food preferences of children with ASD, using data gathered through caregiver-completed questionnaires. Their study identified food items consumed by over 50% of children with ASD. Among fruits and proteins, commonly consumed items included apples, grapes, apple and grape juice, baked chicken, chicken nuggets, hot dogs, ice cream, and peanut butter. Preferred starches among this group included cake, cookies, crackers, French fries, pasta, pizza, potato chips, pretzels, spaghetti, and white bread [55].

Children with Autism Spectrum Disorder typically have a restricted and selective diet. Studies indicate that their diet often includes limited fruits and vegetables. Commonly consumed foods include high-calorie snacks, processed foods, and carbohydrate-rich items such as pasta, pizza, biscuits, and white bread. Dairy intake is reduced with lower milk and yogurt consumption, while dairy-free soft drinks and sweetened fruit juices are frequently preferred. Protein intake is also limited, with lower lean meats and fish consumption. A typical meal might consist of products like chicken nuggets, French fries, or pasta, accompanied by fruit juice, with limited fresh vegetables or varied protein sources (Figure 1).

## 7. Diet Quality of Patients with ASD

A review of the diets of children with autism may raise concerns about their adequate nutrition and the provision of all necessary macro- and micronutrients. Food selectivity, neophobia, and the avoidance of specific foods, coupled with a preference for others, places children with ASD at greater risk of nutritional deficits. Moreover, as parents experience a range of challenges when introducing new foods, their rising anxiety, concern, and frustration frequently result in children being served the same meals repeatedly [8,56]. Parents tend to concentrate on mealtime behavior, rather than reflecting on the appropriate nutritional approaches for their children [57].

In the study mentioned above by Plaza-Diaz et al., the research group was characterized by higher energy density and elevated levels of saturated fats, calcium, and vitamin C, while showing deficiencies in iron, iodine, and B vitamins (such as riboflavin and folic acid) [51]. Consistent findings across various studies indicate additional deficiencies in micronutrients: calcium, zinc, potassium, sodium, magnesium, iron, selenium, and copper [58,59,60]. Vitamins A, C, D, and E deficiencies are also commonly identified [61]. In some cases, severe vitamin C deficiency has even resulted in scurvy [62]. Several studies have indicated that children with ASD exhibit impaired bone development and reduced mineral density, which in turn increases the risk of fractures. This may be associated with reduced consumption of calcium and vitamin D [63].

Feeding disturbances in ASD children may contribute significantly to the risk of malnutrition. Children with ASD are particularly vulnerable to being underweight, often due to reduced caloric intake from selective food rejection [64]. At the same time, they face an increased risk of overweight and obesity, attributed to preferences for calorie-dense foods high in carbohydrates, simple sugars, and fats [65].A study conducted in Turkey revealed that the majority of the sampled children exhibited either overweight or obesity (58.5%). Furthermore, 11% of the children were classified as thin (3rd–15th percentile) or severely thin (<3rd percentile) [59].The results of a study based on US nationally representative data indicate that the prevalence of obesity in children with ASD was 30.4%, compared to 23.6% of children without autism [66]. These dietary patterns may also heighten their susceptibility to metabolic conditions, such as cardiovascular diseases, hypertension, and diabetes [67].

In their study, Demir et al. found that the dietary behavior of children with ASD may affect their anthropometric outcomes, as children with ASD had higher BMI z-scores and lower height z-scores compared to healthy controls [68].

Despite the limited variety of foods consumed and food refusal, most research did not show significant differences in total caloric, carbohydrate, protein, or fat intake between the ASD and typically developing control groups [69].Current literature lacks a definitive consensus on these findings, necessitating further research to achieve greater clarity. A notable limitation of survey-based studies assessing dietary patterns and nutrient intake in children with ASD is their reliance on questionnaire data, which depends heavily on the accuracy and recall of participants regarding food types and quantities consumed. Additionally, these questionnaires are often administered over a short time frame (e.g., one to several days), which may not adequately reflect long-term dietary patterns. Furthermore, many studies focus on cohorts from specific countries, potentially introducing cultural and dietary differences that limit the generalizability of findings to children in other regions.

## 8. Characteristic of Gastrointestinal Symptoms in Children with ASD

Among patients with ASD, gastrointestinal (GI) symptoms are frequently reported as more prevalent and severe. The occurrence of GI symptoms among children with ASD varies widely across studies, ranging from 9% to 84%, compared to 9–37% in children without ASD [70]. The most commonly reported symptoms include abdominal pain and discomfort, constipation, diarrhea, soiling, flatulence, nausea or vomiting, and gastroesophageal reflux (GERD) [71]. Moreover, food allergies are more frequently found in children with ASD, with a rate of 20–25% compared to only 5–8% in children without ASD [72].

Food selectivity and behaviors associated with ARFID impact the gut microbiota composition. Neophobia reduces microbial diversity in the intestines. This decreased diversity is hypothesized to contribute to gastrointestinal symptoms such as abdominal pain and irregular stool consistency, ranging from loose to hard stools [73].

What is more, avoiding large food groups can also affect the gastrointestinal system and exacerbate symptoms like reflux, abdominal pain, nausea, or issues with intestinal motility [74].

A significantly lower intake of fruits and vegetables reduces fiber consumption, and combined with insufficient fluid intake, may hinder healthy intestinal transit and increase the likelihood of constipation [75].

Diarrhea in individuals with ASD may be influenced by dietary preferences, such as excessive carbohydrate and low-fat consumption, as well as overfeeding, food allergies/intolerances, and anxiety or stress. Excessive consumption of sugary drinks or fruit juices may also be a contributing factor, due to the high concentrations of fructose, sorbitol, and mannitol they contain. It should be noted that overflow incontinence in the course of constipation can be mistaken for diarrhea [76]. Pica, which involves eating non-food items, can also increase the risk of diarrhea by irritating the digestive tract, causing bacterial or parasitic infections, and causing nutrient imbalances [77].

It is not fully understood why children with ASD are more likely to have reflux, but it is thought that delayed gastric emptying, impaired esophageal motility, and differences in sensory processing may contribute [19].

Abdominal pain and discomfort in individuals with ASD may stem from a heightened sensitivity to internal sensations, including pressure on the abdomen, the texture of clothing, and other internal stimuli [78,79]. However, any gastrointestinal symptoms in children with ASD should always be considered in the context of common gastrointestinal disorders in the pediatric population.

Some drugs can also impact GI tract function; for example, stimulants can cause abdominal pain, while beta-blockers can result in constipation, diarrhea, and stomach irritation [80,81]. Antipsychotics and antiepileptic drugs can also cause constipation [82].

Because GI symptoms affect children’s ability to concentrate and cause anxiety, they can interfere with development and learning skills. Treating these symptoms can improve behavior, cognitive function, school performance, and overall well-being. Clinicians should consider that behaviors perceived as “picky” may be protective responses. Children with ASD, who often struggle to communicate symptoms, require closer observation. Parents and clinicians must recognize vocal, sensory, and motor cues that may signal GI-related pain [83]. Vocal indicators may include throat clearing, guttural sounds, screaming, moaning, or sobbing. Motor behaviors might include pressing the abdomen, pointing, certain repetitive and unusual movements, abnormal posture, or aggressive and self-injurious actions [84]. GI problems can also cause frequent swallowing, spitting, ear rubbing, and coughing and contribute to sleep problems [85] (Figure 2).

The prevalence of GI symptoms in children with ASD shows considerable variability between studies, most likely due to differences in study populations, methodologies, and diagnostic criteria. Verbal communication difficulties are also a significant barrier to collecting reliable data. Further research is also required for a better understanding of the relationship between the increased incidence of gastrointestinal symptoms in patients diagnosed with autism and the gut–microbiome–brain axis.

## 9. The Gut–Brain Axis and Gut Microbiota in ASD

The gut–brain axis is a bidirectional communication network linking the central nervous system (CNS) with the enteric nervous system (ENS). It plays a crucial role in maintaining homeostasis and influences many processes, including behavior, mood, and cognition [87].

The trillions of microorganisms living in our gastrointestinal tract, elegantly named the gut microbiota, have garnered enormous interest from the scientific world this past decade. This complex system has become a crucial area of study in understanding various neurodevelopmental disorders (NDDs), including ASD (Figure 3).

Gastrointestinal disturbances affect up to 70% of children with ASD, making them one of the most prevalent problems of this condition [100]. Moreover, the severity of these symptoms correlates with the severity of ASD [101,102]. This implies a potential role of the gut microbiota in ASD. The composition and function of gut microbiota are shaped by multiple factors, including mode of delivery at birth, genetic predisposition, physical activity levels, dietary patterns, environmental exposures, infections, and antibiotic usage [103].

Research findings on the characteristic gut microbiota composition in individuals with ASD remain inconsistent, with studies often reporting contradictory results regarding specific bacterial species. Scientists use varying approaches to analyze the composition of gut microbiota. Culture-dependent methods isolate and identify viable bacterial species. Real-time PCR quantifies specific bacterial taxa through DNA amplification and detection. Fluorescence in situ hybridization (FISH) allows for visualization and identification of specific bacterial populations using fluorescently labeled oligonucleotide probes targeting ribosomal RNA. 16S rRNA gene pyrosequencing provides culture-independent, comprehensive profiling of the bacterial communities through high-throughput sequencing of the conserved 16S rRNA gene regions, enabling taxonomic classification of the diverse bacterial populations present in the samples [104,105].

In their meta-analysis, Xu et al. analyzed 9 studies to summarize previous literature on the topic of gut microbiota in ASD [106]. They concluded that participants with ASD had lower amounts of *Akkermansia*, *Bacteroides*, *Bifidobacterium*, *Escherichia coli*, and *Enterococcus* and higher quantities of *Faecalibacterium* and *Lactobacillus*. There was also a slight increase in the abundance of *Ruminococcus* and *Clostridium* [106]. A larger meta-analysis examining 18 studies revealed some contrasting findings. While this study confirmed the increased abundance of *Faecalibacterium* and decreased levels of *Bifidobacterium* in ASD children, it showed higher levels of *Bacteroides*—contrary to Xu et al.’s findings. The study also identified three predominant bacterial phyla in ASD children (Bacteroidetes, Firmicutes, and Actinobacteria) and found increased levels of *Parabacteroides*, *Clostridium*, and *Phascolarctobacterium*, along with reduced *Coprococcus* abundance [107].

Desbonnet et al. documented that germ-free (GF) mice prefer to spend more time in isolation rather than interacting with other mice. They also showed that colonizing these mice with pathogen-free gut microbiota increased sociability but did not affect social novelty [108]. Some studies show that mice with gut microbiota depleted by antibiotics also show disorders in social behavior [109,110]. Offspring of mice that were fed with a high-fat diet (HFD) during pregnancy showed ASD-like social symptoms. Introducing certain bacteria, such as *L. reuteri*, improved social deficits in these mice [110]. Improvement of social behavior was correlated with increased levels of oxytocin in the hypothalamus, a hormone crucial for the modulation of social interactions [111]. The vagus nerve is a prerequisite for the effectiveness of reversing social deficits by *L. reuteri*, as vagotomized mice did not show improvements in social behavior after introducing *L. reuteri* [112].

Bove et al. demonstrated a potential role of chemical signaling in a BTBR mice model. BTBR mice are an inbred mouse strain that present human autism-like behaviors [113]. Analyses of serum and intestinal contents showed decreased amounts of 5-amino valeric acid (5AV) and taurine, weak gamma-aminobutyric acid (GABA) receptor agonists. Administering 5AV and taurine to BTBR mice improved their social behavior and reduced repetitive behaviors [114].

The immune pathway’s role in linking gut microbiota and the brain is clearly outlined in a model of ASD induced by maternal polyinosinic:polycytidylic acid (poly(I:C)) [115]. Poly(I:C) is a synthetic analog of viral double-stranded RNA (dsRNA) that triggers the typical inflammatory response seen during viral infections [116]. It is a crucial part of the maternal immune activation (MIA) hypothesis that proposes disturbance of fetal neurodevelopment in response to inflammatory reaction [117]. In offspring of mice treated with poly(I:C), there are alterations in the composition of the gut microbiota, as well as increased levels of the microbial metabolite 4-ethylphenylsulfate (4EPS), which can induce anxiety-like behaviors [115]. Supplementation of MIA offspring with the human commensal *Bacteroides fragilis* corrected defects in gut permeability, reduced anxiety-like and stereotypic behavior, as well as decreased levels of 4EPS [115]. These findings further confirm the role of the microbiota–gut–brain axis in ASD.

## 10. How ARFID, Dietary Selectivity, and Dietary Neophobia Can Affect the Gut Microbiome?

Despite the topic’s importance, research remains limited, with just one study by Ye et al. addressing the topic of ARFID and its impact on the gut microbiota [36]. Their study included 102 children with ARFID and 33 healthy children (HC). The researchers analyzed stool samples using 16S rDNA and metagenomic sequencing. 16S rDNA sequencing targets the conserved 16S ribosomal RNA gene to enable bacterial taxonomic classification based on hypervariable regions. Shotgun metagenomic sequencing sequences all microbial DNA present in a sample without target amplification [104]. While 16S rDNA sequencing allows for bacteria identification only to the genus level, metagenome sequencing allows for more precise information, such as information at the species level. Ye and colleagues found that the richness of microbial species (Chao1 index) was greater in healthy children, while the diversity of microbial species (Shannon index and Simpson index) was higher in the ARFID group. The microbial composition differed between groups, with the HC group characterized by enrichment of Actinobacteriota and its hierarchical taxa (class Actinobacteria, order Bifidobacteriales, family Bifidobacteriaceae, and genus *Bifidobacterium*). Conversely, the ARFID group showed elevated levels of both Enterobacterales (including family Enterobacteriaceae) and Bacteroidaceae (including genus *Bacteroides* and species *B. vulgatus*) [36] (Figure 4).

Analysis of the ARFID group revealed alterations in their gut bacterial composition. These changes followed two concerning patterns: an increase in potentially harmful bacteria and a decrease in beneficial ones. The increased bacteria included members of the Enterobacteriaceae family, which are well-known for their pathogenic associations [118]. Another increased species was *Bacteroides vulgatus*, which shows complex effects on gut health. While *Bacteroides* species generally benefit gut health by producing short-chain fatty acids (SCFAs) [119], *B. vulgatus* specifically has shown contradictory effects: it can both reduce inflammation in some conditions [120] and promote inflammatory bowel disorders in others [121]. Concurrent with these increases, ARFID patients showed markedly reduced levels of beneficial *Bifidobacterium* species. Given *Bifidobacterium’s* established role in immune regulation, inflammation control, and potential anti-aging effects [122], this bacterial profile suggests a dysbiotic state that may contribute to ARFID pathogenesis.

## 11. The Gut–Brain–Behavior Cascade: A Self-Perpetuating Model in Autism Spectrum Disorder

When considering all the aspects mentioned above, a compelling pattern emerges. Research shows that individuals with ASD experience FS and ARFID significantly more often than the NT population. Their selective and limited diet reduces gut microbiota diversity [123,124]. Furthermore, an imbalanced gut microbial flora can intensify autistic behaviors. This creates a cycle—ASD symptoms can contribute to the development of ARFID, which leads to microbiota degradation, further exacerbating ASD symptoms and creating a self-reinforcing loop. (Figure 5).

The gut–brain axis is a bidirectional communication system connecting the gastrointestinal tract and central nervous system through neural, endocrine, and immune pathways. Recent research shows that gut health significantly influences brain function and behavior, particularly in neurodevelopmental conditions like ASD [128].

The gastrointestinal tract contains its own nervous system called the enteric nervous system (ENS) [129,130], often known as the “second brain”, which communicates with the brain primarily through the vagus nerve. The ENS consists of millions of neurons that control digestive functions [131], while the vagus nerve provides bidirectional communication through sensory and motor fibers [88], allowing the gut and brain to exchange information about mechanical changes, hormone levels, and various other physiological signals [132,133]. Gut dysbiosis may worsen or induce ASD symptoms through multiple biological mechanisms and metabolic pathways.

The endocrine pathway connects ASD and the gut–brain axis through hormone secretion from enteroendocrine cells (EECs) [134,135]. These cells produce serotonin, glucagon-like peptide-1 (GLP-1), and peptide YY (PYY), affecting brain function and behavior [89,90,136]. While gut-produced serotonin cannot cross the blood–brain barrier, it influences mood and social behavior through peripheral systems [136,137,138,139,140]. The gut microbiota affects serotonin production by regulating tryptophan hydroxylase 1 (TPH1) enzyme. GLP-1 and PYY signal the brain via the vagus nerve, impacting appetite and feeding [89]. Disrupted endocrine signaling from altered gut microbiota may influence ASD behaviors and feeding patterns.

The hypothalamic–pituitary–adrenal (HPA) axis, another key pathway, is affected by gut microbiota changes [91]. Dysbiosis leads to heightened stress responses with elevated cortisol and adrenocorticotropic hormone (ACTH) levels [141,142]. Research with germ-free mice shows that lacking gut microbiota causes exaggerated HPA stress responses [143]. This chronic HPA activation can worsen anxiety, cognitive function, and stress responses in ASD individuals.

Furthermore, altered gut microbiota can lead to immune system modulation resulting in neuroinflammation [92]. The gut microbiota influences the development and function of microglia, the brain’s resident immune cells [93]. Dysbiosis leads to decreased maturity and function of microglia in a mouse model [144], contributing to neuroinflammation associated with ASD [98].

Microbial metabolites provide another gut–brain communication pathway. SCFAs produced by gut bacteria are the most often mentioned in the context of ASD.

High propionate levels correlate with ASD-like behaviors in rodent studies, affecting neural organization and behavior [145]. On the contrary, increasing butyrate levels may improve social interactions and decrease repetitive behaviors [146]. Sodium butyrate additionally decreases neuroinflammation [147,148].

SCFAs additionally influence neuroplasticity and neurogenesis by modulating neurotrophic factors, including brain-derived neurotrophic factor (BDNF), nerve growth factor (NGF), and glial cell line-derived neurotrophic factor (GDNF) [149,150].

SCFAs also maintain blood–brain barrier (BBB) integrity through tight-junction protein regulation, as demonstrated in studies with germ-free mice [94,95,96].

GF mice present numerous social deficits, like reluctance towards new relationships and avoiding contact with unknown mice. They spend significantly more time on repetitive behaviors like self-grooming. Hence, they are often treated as ASD mouse models and used to assess intestinal microbiome impact on behavior [124].

Research on them [114] shows that transplanting fecal microflora from ASD donors to GF mice induced autistic behavior, specifically increased repetitive behavior, decreased locomotion, and communication, compared to mice with NT donors microbiota. This is further confirmed by Xiao et al. [151], who performed a similar experiment, where mice seeded with ASD microbiota presented more autistic-like behaviors.

In a mouse model of ASD induced by maternal Poly(I:C), alterations in gut microbiota composition and increased levels of the microbial 4EPS are observed in offspring, which can induce anxiety-like behaviors. Supplementing these offspring with the human commensal *Bacteroides fragilis* improved gut permeability, reduced anxiety-like and repetitive behaviors, and decreased 4EPS levels [115]. The effects of gut microbiota on eating behavior and ASD symptoms are summarized in Table 2.

This self-perpetuating cycle highlights the deep connection between feeding disorders, gut health, and ASD symptoms. While no direct research has confirmed this cyclic relationship in its entirety, analysis of each individual step reveals this pattern. Further research is needed to fully understand these connections and to confirm the findings from animal models. There is a lack of specific evidence for alterations in sensory–sensitivity being modulated by gut microbiota, but there are multiple mechanisms allowing the gut microbiome to influence the host’s brain development and function.

Breaking this cycle requires a holistic approach with interventions targeting each stage. To address ASD symptoms that contribute to ARFID, behavioral strategies can be employed. To manage ARFID and its effects on the microbiota, implementing specific dietary recommendations can help ensure balanced nutrition and support gut health. Restoring microbiota balance and alleviating ASD symptoms may involve the use of probiotics and regular physical activity. Together, these interventions aim to improve outcomes and overall well-being for individuals affected by both conditions.

## 12. Clinical Implications and Future Research Directions

Research suggests a complex relationship between ASD, selective eating disorders like ARFID, and gut microbiota. Sensory oversensitivity in individuals with ASD often leads to restrictive diets, which can reduce gut microbiota diversity. This, in turn, may impact brain function through the gut–brain–microbiota axis, creating a self-perpetuating cycle of gastrointestinal and behavioral challenges.

To address these challenges, multidisciplinary therapeutic approaches that emphasize nutrition could improve both gastrointestinal and behavioral symptoms in individuals with ASD.

A multidisciplinary approach should prioritize psycho-educational and therapeutic activities, such as behavioral strategies, that engage both the patient and their parents. Behavioral treatment strategies are based on the combination of adjustments to the mealtime setup, attitude management, and caregiver training. Behavior management procedures are planned to enhance adaptive behaviors and decrease maladaptive behaviors [166]. A commonly used method is also a simple mixing of preferred and non-preferred food [154]. The goal of all these programs is to reduce challenging behaviors and expand the variety and quantity of foods children accept, minimizing nutritional risks. Ismail et al. emphasize that parents play a crucial role in ensuring their children receive sufficient nutrition [167]. Parental perceptions of their child’s nutritional status, particularly diet quality, are shaped by a variety of social, biological, economic, and psychological factors. Parents who realize and indicate concern about their child’s unhealthy weight are often more motivated to improve the family’s diet and effectively manage the child’s eating and physical activity habits. Tan et al. highlight that parental strategies to enhance food acceptance in children typically involve offering a variety of meals at home, regularly introducing new and similar foods, providing positive reinforcement, and modeling healthy eating behaviors [168]. However, practices such as controlling food intake, pressuring the child to eat, restricting portions, or using rewards can negatively impact mealtime behavior. The study also suggests that parents often experience heightened emotional responses during mealtimes, which can influence how they interact with their children.

In the future, a multidisciplinary approach could consider incorporating dietary and supplementation-based treatments. However, evidence supporting the effectiveness of specific diets, such as the anti-inflammatory or ketogenic diets, is currently limited.

In the case of the anti-inflammatory diet, its potential relevance stems from the elevated levels of oxidative stress and inflammation observed in individuals with ASD, which are thought to play a key role in the disorder’s pathogenesis [169]. These processes are closely interconnected, as inflammatory responses can induce oxidative stress and mitochondrial dysfunction. This interaction can amplify oxidative damage, creating negative feedback that disrupts brain development and functioning and potentially contributes to the development of neurodevelopmental disorders like ASD. Plant-based antioxidants such as carotenoids, phenolics, flavonoids, and vitamins are known for their anti-inflammatory, antioxidant, and immune-boosting properties. Thus, an antioxidant-rich diet can help reduce issues associated with ASD by decreasing the harmful effects of Reactive Oxygen Species. These nutrients may support neurological and gastrointestinal health, helping to improve food tolerance and potentially alleviate sensory sensitivities and food neophobia [170].

The ketogenic diet (KD), on the other hand, has limited evidence suggesting it may reduce bacterial dysbiosis and promote BDNF levels, as noted by Allan et al. [171]. They observed a reduction in the expression of pro-inflammatory cytokines, particularly IL-1β and IL-12p70, in the plasma of ASD patients after following the KD. They also noted a significant reduction in plasma levels of BDNF, a key factor involved in neuroinflammation in the central nervous system that has been previously associated with ASD. The KD demonstrated a positive effect on gut microbiota, enhancing diversity and promoting a more balanced composition. These results indicate improved gut health and a reduction in the pathogenic potential of gut microbes. During KD, there is an increased level of ketone bodies, like β-hydroxybutyrate, which can cross the BBB and may have neuroprotective effects resulting in behavioral improvement in children with ASD. The precise mechanism by which gut flora interact with liver-derived ketone bodies remains unclear. Interestingly, the gut-derived short-chain fatty acid butyrate shares structural similarities with β-hydroxybutyrate and has been proposed as a potential supplement for ASD. However, the effectiveness of such a regimen raises concerns, particularly in cases of increased ARFID, which may result in nutritional deficiencies. A strict KD would require careful oversight by a dietitian to ensure adequate intake of nutrients like iron and amino acids, preventing potential worsening of the condition in ASD patients due to non-adherence. Additionally, there is insufficient evidence on the long-term effects of this diet, and no current studies conclusively support its recommendation. The KD can cause side effects [172]. Most of them are manageable with careful monitoring, particularly in the initial weeks. Common issues include hypoglycemia, dyslipidemia, gastrointestinal symptoms, carnitine deficiency, bone conditions (e.g., osteopenia, osteoporosis), nephrolithiasis, and growth delays, requiring close supervision in pediatric cases. Rare complications, such as pancreatitis, prolonged QT intervals, and transient vascular changes, have also been reported. Long-term studies in children with epilepsy on KD highlight potential risks to bone health, including increased fractures and reduced bone mineral density.

Given the role of the microbiota in ASD, therapeutic approaches extend beyond dietary interventions. Dysbiosis of the microbiota can also be directly targeted with probiotics. Probiotic intake has been shown to benefit health by promoting positive changes in gut microbiota, and studies suggest that probiotic interventions in children with ASD may be a valuable complementary therapy for symptom relief [173,174]. Kwak et al. suggest that modulating the gut microbiome with psychobiotics—probiotics known to positively affect neurological function and hold potential for treating psychiatric conditions—offers a promising therapeutic approach for ASD [175]. Psychobiotic strains commonly belong to the *Lactobacilli* and *Bifidobacteria* families. In ASD, the gut microbial composition is significantly altered, with differences in the Firmicutes-to-Bacteroidetes ratio and reduced levels of *Fusobacteria* and *Verrucomicrobia* compared to neurotypical children. These microbial communities are integral to the production of substances such as SCFAs, which have key neurobiological roles in the microbiota–gut–brain axis, and 4-ethylphenylsulfate, a tyrosine metabolite implicated in ASD-like behaviors. Liu et al.’s analysis reveals probiotic interventions with *Lactobacillus plantarum* have been explored in the Taiwanese population. The Taiwanese study involved a one-month treatment with over 70 children and pre-adolescents, showing some symptom improvements, though none were statistically significant. Longer treatments, however, demonstrated significant benefits in reducing irritability, stereotyped behaviors, and hyperactivity. Despite these promising results, the studies were limited by small sample sizes and a lack of analysis of baseline gut microbiota composition [176]. The main limitation of the impact of probiotics is that there are not many studies available on the impact of long-term supplementation. In addition, the risks involved have not been clearly proven so far. Although research is limited, probiotics are suspected to have a beneficial effect in alleviating gastrointestinal symptoms in individuals with ASD.

Such a small amount of evidence does not allow the transfer, at this point, of the realities of research to the daily lives of patients. From a nutrition and diet perspective, attention should also be paid to the potential role of physical activity, which occupies an essential place in the nutritional pyramid. Regular physical activity is essential for individuals with ASD as it not only enhances physical health but also improves behavioral outcomes [177]. Due to a higher prevalence of sedentary lifestyles among this population, which contributes to obesity and related health issues, promoting active lifestyles is crucial. Both children and adults with ASD often encounter challenges such as difficulties in social interaction, communication impairments, and a tendency towards inactivity. Engaging in regular physical exercise can help alleviate these challenges by enhancing motor skills, improving social interactions, and reducing anxiety and behavioral problems [178]. Research has shown that activities like swimming, horseback riding, and structured group exercises significantly improve social skills, communication, and motor coordination [179]. Additionally, exercise can decrease problem behaviors such as repetitive actions and self-injury by promoting relaxation and increasing the release of neurotransmitters like endorphins and dopamine, which enhance brain function [180]. Incorporating physical activity into the daily routine of individuals with ASD requires understanding their preferences and providing appropriate motivation. Tailoring activities to individual interests can increase participation and enjoyment, thereby maximizing the benefits of exercise [181].

Overall, there is increasing evidence supporting the role of dietary interventions in improving gastrointestinal symptoms in children with ASD. However, more research is needed before such data can be used as formal guidelines. New studies, ideally randomized or cross-sectional, should focus on children with similar, preferably significant, ARFID and subject them to the described interventions. Studies should include different age groups to capture developmental differences; gender-specific analyses are also recommended due to potential gender-dependent differences in microbiota composition. Methodologically, research should employ advanced techniques like 16S rRNA sequencing and metagenomics while controlling for confounding factors such as baseline diet or medication use. Practical challenges such as parental adherence, cultural dietary practices, and socioeconomic constraints must be addressed through caregiver training and psychoeducational programs. Additionally, the long-term impact of these interventions must be evaluated, alongside the monitoring of peripheral blood concentrations of vitamins, minerals, inflammatory markers, and bone status through X-rays. Even with such studies, challenges related to the nature of ASD and research rigor will remain. Therefore, these studies should be conducted over an extended period, at least one year, and involve multiple centers.

## 13. Limitations and Strengths

Our review faces several limitations, most notably the lack of a systematic approach, which impacts the robustness of the findings. This is a relatively new and emerging field of research, and as such, the limitations often stem from the exploratory nature of the studies conducted so far. A key issue is the heterogeneity of the data, with many studies involving small sample sizes, focusing on specific populations (e.g., single ethnicities or regions), or relying heavily on parental reports, which can introduce subjective bias and reduce the generalizability of the results. Additionally, much of the mechanistic understanding is derived from animal models, which may not fully translate to human conditions. Inconsistent findings regarding gut microbiota composition in individuals with ASD further complicate interpretations, reflecting methodological differences across studies. The predominance of cross-sectional research, rather than longitudinal studies, also limits our ability to establish temporal relationships or causality between ASD symptoms, ARFID, and gut dysbiosis. Evidence for therapeutic interventions, including dietary modifications and probiotics, remains limited, with the long-term effects of approaches such as ketogenic or anti-inflammatory diets still unclear. Moreover, many proposed mechanisms, particularly those linking sensory sensitivities in ASD to microbiota alterations, lack robust validation. The variability in microbiota analysis techniques and reliance on parental observations further introduce biases related to family dynamics and reporting accuracy, hindering comparability and synthesis of results. To establish definitive causal relationships within the proposed cycle and determine the impact of specific interventions, further research, particularly with more standardized methodologies and larger, more diverse sample sizes, is essential.

The strengths of this study lie in its holistic approach, offering a thorough exploration of the complex relationships between ASD, eating challenges, gastrointestinal symptoms, and gut microbiota. By integrating these diverse aspects, the paper provides a valuable and insightful perspective on how they are interconnected. A significant strength is the introduction of a novel conceptual framework, which illustrates the bidirectional relationships where ASD symptoms worsen ARFID, leading to changes in gut microbiota that, in turn, exacerbate ASD symptoms. The study also offers valuable therapeutic insights, discussing a variety of strategies, including behavioral interventions, parental involvement, dietary modifications, and the potential of emerging treatments like probiotics and psychobiotics. Because of that, it offers practical recommendations for multidisciplinary approaches to improve outcomes for individuals with ASD. Additionally, the study seeks to identify important gaps in current research, suggesting key areas for future investigation.

While our review focuses on exploring dietary patterns, gastrointestinal symptoms, and the composition of the gut microbiota in children diagnosed with ASD, Schneider et al. [182] have addressed a related topic within the context of ARFID. Their conceptual model of the microbiota–gut–brain axis provides valuable insights into how restrictive eating behaviors can influence gut microbiota diversity and, in turn, affect psychological and physiological processes.

Both reviews highlight the significant role of dietary selectivity in shaping gut microbiota and its potential implications for gastrointestinal and behavioral symptoms. However, ASD-specific factors, such as heightened sensory sensitivities, distinguish the challenges faced by this population. These sensitivities often lead to selective eating patterns that exacerbate nutritional deficiencies and uniquely impact gut microbiota composition. We observe some shared conclusions by comparing findings, particularly regarding the bidirectional relationship between diet and the gut–brain axis. Nonetheless, our review places greater emphasis on how the specific characteristics of ASD, including sensory-related food selectivity, contribute to these interactions, aiming to complement and build upon the insights provided by Schneider et al. [182].

## 14. Conclusions

ASD is closely linked to dietary selectivity, gastrointestinal dysfunction, and gut microbiota alterations, forming a self-perpetuating cycle that impacts both physical and behavioral health. The interplay between ASD symptoms, ARFID, and gut dysbiosis highlights the crucial role of the gut–brain axis in neurodevelopmental disorders. While evidence suggests that dietary interventions, probiotics, and behavioral strategies can improve symptoms, further research is needed to establish standardized therapeutic approaches. A multidisciplinary strategy addressing both dietary and gut health is essential for optimizing outcomes and improving the quality of life for individuals with ASD.

## Figures and Tables

**Figure 1 nutrients-17-00486-f001:**
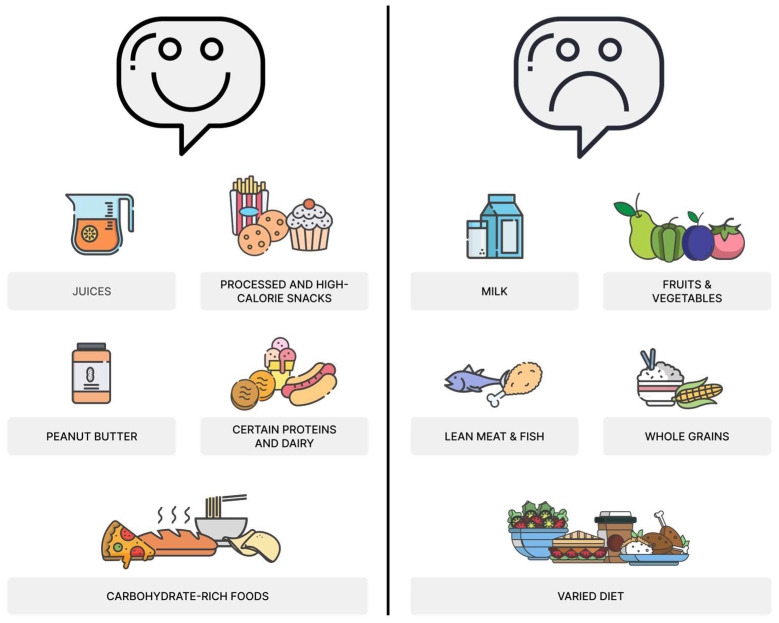
Children with autism spectrum disorder (ASD) often display selective eating habits, favoring high-calorie snacks, processed foods, and refined carbohydrates like pasta, pizza, and white bread while consuming fewer fruits, vegetables, lean meats, fish, and whole grains. Dairy intake is typically low, while sweetened beverages and juices are preferred. These dietary patterns increase the risk of nutritional deficiencies, including calcium, iron, iodine, zinc, and vitamins A, D, E, and B complex. Calcium and vitamin D deficiencies can impair bone development, while a preference for calorie-dense foods raises the risk of obesity and metabolic conditions like diabetes and hypertension.

**Figure 2 nutrients-17-00486-f002:**
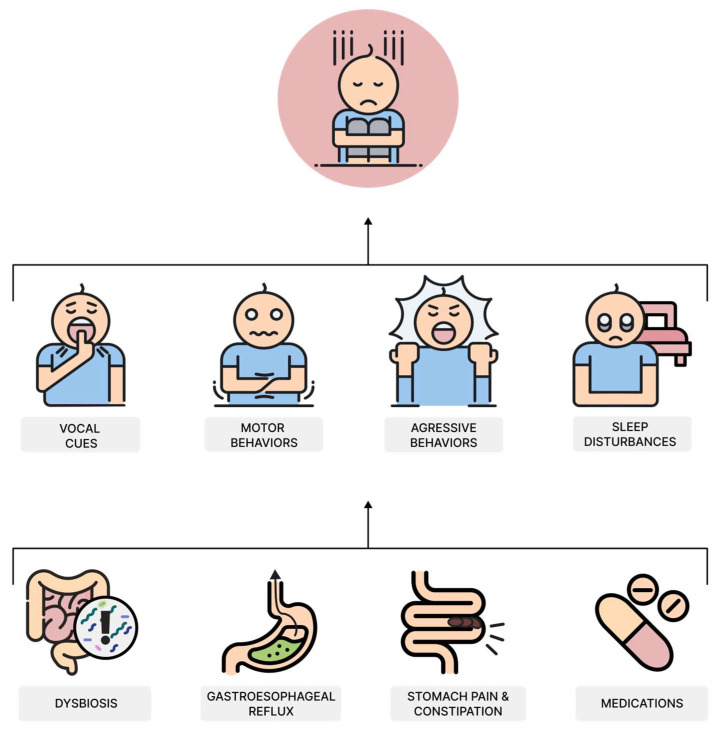
Children with autism spectrum disorder (ASD) often experience gastrointestinal (GI) symptoms, which are communicated through non-verbal cues due to their difficulty in expressing discomfort. These cues can be categorized into vocal behaviors (e.g., throat clearing, moaning, sobbing), motor behaviors (e.g., abdominal pressing, pointing, self-injury), aggressive behaviors (e.g., irritability, tantrums), and sleep disturbances (e.g., difficulty falling or staying asleep) [83,84]. Common causes include dysbiosis, gastroesophageal reflux, stomach pain, constipation, and medication side effects [72,82,86]. Early identification of these behaviors is crucial for timely intervention, improving the child’s overall well-being.

**Figure 3 nutrients-17-00486-f003:**
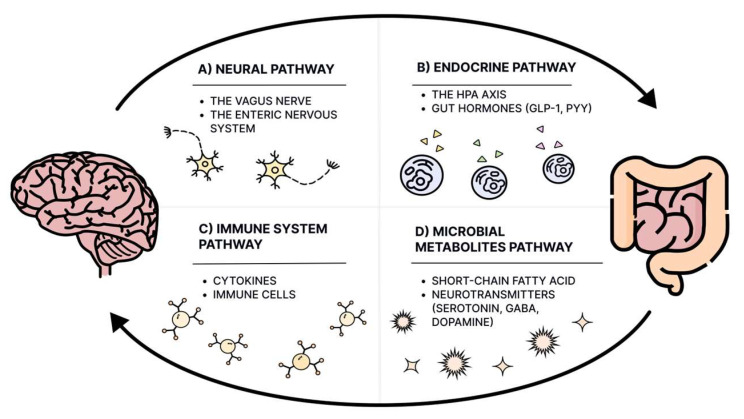
The gut–brain axis is a complex, bidirectional communication network linking the central nervous system (CNS) and the enteric nervous system (ENS), playing a key role in regulating mood, cognition, and overall health. This interaction occurs through four primary pathways: (**A**) the neural pathway, where the vagus nerve facilitates direct communication between the gut and brain [88]; (**B**) the endocrine pathway, involving the hypothalamic–pituitary–adrenal (HPA) axis and gut hormones like GLP-1 and PYY to mediate stress responses and regulate appetite [89,90,91]; (**C**) the immune system pathway, where cytokines and immune cells modulate neuroinflammation and brain function [92,93]; (**D**) the microbial metabolites pathway, where microbial byproducts like short-chain fatty acids (SCFAs) influence neurotransmitter production and maintain blood–brain barrier integrity [94,95,96,97]. Disruptions in these pathways, such as dysbiosis or increased intestinal permeability, can lead to neuroinflammation, altered neurotransmitter levels, and various neurological or psychiatric disorders [98,99].

**Figure 4 nutrients-17-00486-f004:**
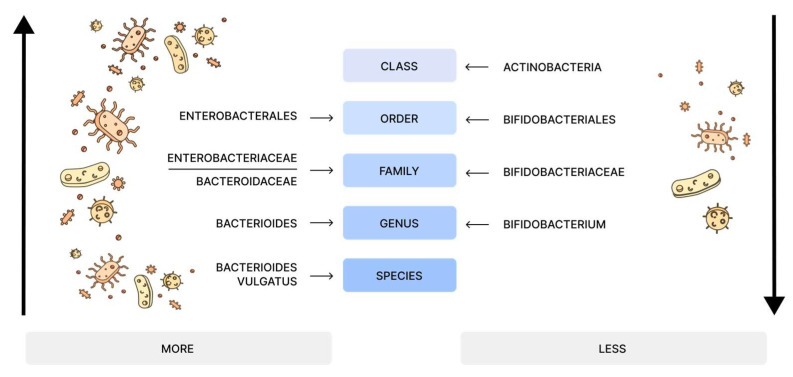
Children with ARFID exhibited higher microbial diversity but lower species richness compared to healthy children. Their gut microbiota showed increased levels of potentially harmful bacteria, including Enterobacteriaceae and *Bacteroides vulgatus*, which have mixed effects on gut health—both reducing inflammation in some contexts and promoting inflammatory bowel disorders in others [36,118,119,120,121]. In contrast, beneficial *Bifidobacterium* species, known for supporting gut health, were significantly reduced in the ARFID group [122].

**Figure 5 nutrients-17-00486-f005:**
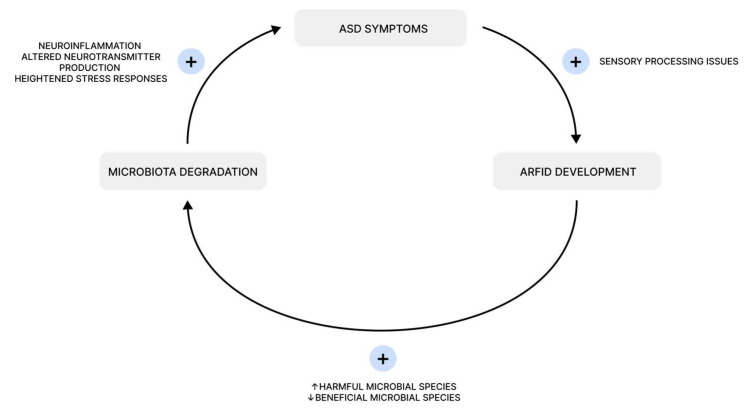
A self-perpetuating cycle links ASD symptoms, ARFID development, and gut microbiota degradation. Sensory processing issues in ASD, such as atypical sensory modulation and oral over-sensitivity, drive food selectivity and neophobia, increasing ARFID prevalence [39]. ARFID disrupts gut microbiota, reducing beneficial species like *Bifidobacterium* and increasing harmful bacteria like *Enterobacteriaceae* and *Bacteroides vulgatus* [36]. This dysbiosis affects the gut–brain axis through neural, endocrine, immune, and metabolic pathways, exacerbating ASD symptoms via neuroinflammation, altered neurotransmitter production, and heightened stress responses [97,98,125,126,127].

**Table 1 nutrients-17-00486-t001:** Subtypes of avoidant/restrictive food intake disorder (ARFID) [32,33,34].

ARFID Subtype	Description	Key Characteristics	General Prevalence
Lack of Interest	Characterized by a lack of appetite or interest in eating or food.	Small bites Slow eating Prolonged mealtimes Chronic low appetite No significant comorbidity with ASD	25.1% [32] 57.6% [33]
Sensory Sensitivity	Avoidance of foods due to sensory issues, such as taste, smell, texture, or appearance.	Rigid eating behaviors Food selectivity and neophobia High comorbidity with ASD [34]	29.5% [32] 21.2% [33]
Fear/Aversive	Avoidance of food due to fear of aversive consequences, such as choking, vomiting, or pain.	High anxiety levels Often triggered by traumatic events (e.g., choking episode) Shorter symptoms duration High comorbidity with anxiety disorders	7.2% [32] 9.1% [33]
Combined Presentation	Avoidance of food due to fear of aversive consequences, such as choking, vomiting, or pain.	High probability of all symptoms except fear-based avoidance Younger age at diagnosis High comorbidity with ASD and learning disabilities	38.2% [32]

**Table 2 nutrients-17-00486-t002:** The effects of gut microbiota on eating behavior and ASD symptoms.

Mechanism	Impact of Gut Microbiota on Eating Behavior/ASD Symptoms	Key Findings
Social behavior and communication modulation	Microbiota influences social interactions and communication behaviors through neural pathways	Behavioral improvements were seen through microbiota modulation [110,112] *Lactobacillus reuteri* supplementation improved social behaviors via vagal signaling [112] It also affects oxytocin-dopamine pathways in the brain [110,112]
Serotonin regulation	Microbiota affects appetite and satiety Influences mood and behavior	ASD patients show hyperserotonemia [152,153,154] Altered serotonin metabolism in both the gut and brain affects gastrointestinal motility and mood [153,155,156]
Dietary diversity impact	Microbiota affects food preferences	A less diverse diet correlates with lower microbiota diversity [157] Altered gut microbiota in ASD patients may simply result from dietary preferences and not ASD itself [157]
Metabolite production	Bacterial metabolites affect behavior They can influence anxiety and irritability	4-EPS linked to anxiety [158,159] Clinical improvements are seen when targeting these metabolites [158]
Gastrointestinal function	Microbiota affects gastrointestinal system functions	GI symptoms are common in ASD (constipation, diarrhea) [160,161,162]
Inflammation and immune response	Inflammatory state affects appetite, influences gut–brain communication, and impacts behavior	Specific bacteria linked to inflammatory markers [163] Correlation of ASD with pro-inflammatory cytokines [163]
Early development	Microbiota shapes future eating patterns, influences neurodevelopment, and affects long-term behavior	Maternal microbiome affects offspring development [110,112,164] Early-life disruption has lasting effects [110,112,164]
Intestinal epithelium barrier function	Microbiota affects nutrient absorption, influences gut–brain communication, and impacts food tolerance	Altered intestinal barrier integrity in ASD, also known as “Leaky gut” [115,165]

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
