# Peer review of "Unraveling the Connections: Eating Issues, Microbiome, and Gastrointestinal Symptoms in Autism Spectrum Disorder"

_nutrients, 2025, doi:10.3390/nu17030486_

Round 1

Reviewer 1 Report

Comments and Suggestions for Authors

Very interesting publication. Quite extensively written, exhaustive. Although I got a bit tired reading it all, it is interesting. The illustrations are also nice to look at. I will remember your article, I would like to quote it in the future:)

However, in many places there is no citation. Unless it is a summary of some point? But for example in paragraph 222-229 there is no citation. Please complete, e.g. by starting the paragraph/sentence with "to sum up" or add a citation, because I do not know whether you are writing about some study or if it is your own thoughts. If there is no citation, it cannot be that there are several lines of text and zero citations in the text or at the end of the sentence/paragraph.

"2. Materials and Methods" please provide information in the inclusion criteria that you were looking for publications, e.g. original studies involving children. Your target group in the review is children from what I understand.

"databases to identify relevant peer-reviewed articles published up to December 2024." Were articles taken from the last 10 years, for example, or from the beginning of the database? It is worth adding this information.

line 588: no period after "139] "

line 887: no period at the end of the sentence.

Author Response

Dear Reviewer,

Thank you for your valuable feedback. We appreciate your interest in our work and your insightful comments. While the manuscript is indeed extensive, our intention was to provide a comprehensive overview of the topic in a single paper. Below, we address your comments in detail:

C1: However, in many places there is no citation. Unless it is a summary of some point? But for example in paragraph 222-229 there is no citation. Please complete, e.g. by starting the paragraph/sentence with "to sum up" or add a citation, because I do not know whether you are writing about some study or if it is your own thoughts. If there is no citation, it cannot be that there are several lines of text and zero citations in the text or at the end of the sentence/paragraph.

R1: Thank you for this suggestion. To improve readability, we have now included brief summaries at the end of each chapter (except for Chapters 3 and 4, which serve as introductions to the topic). These summaries, now highlighted, provide a general overview of the content, making it easier to navigate the manuscript if it appears too extensive. Additionally, we have carefully reviewed the entire manuscript and ensured that all previously missing citations have now been included.

C2: "2. Materials and Methods" please provide information in the inclusion criteria that you were looking for publications, e.g. original studies involving children. Your target group in the review is children from what I understand. 
R2: Thank you for this observation. We have revised the methodology section for clarity. While our primary focus is on original research involving children, we do not exclude relevant studies on the gut microbiota, gut-brain axis, that obtain information from animal studies. This approach, along with its rationale, has been explicitly stated in the revised section.

C3: "databases to identify relevant peer-reviewed articles published up to December 2024." Were articles taken from the last 10 years, for example, or from the beginning of the database? It is worth adding this information."
R3: We have now included additional details to better illustrate our search strategy and enhance transparency.

C4 and C5: 

line 588: no period after "139] "

line 887: no period at the end of the sentence.

R5: We appreciate your attention to detail. The indicated sections have been carefully revised and corrected accordingly.

All changes, including comments from others Reviewers, should be available at word file. Please see the attachment. 

Best regards,

Stefan Modzelewski and coauthors

Reviewer 2 Report

Comments and Suggestions for Authors

Dear Authors,

The review of the submitted article highlights several critical elements that require significant improvement. The paper addresses an important topic regarding the interactions between eating issues, the microbiome, and gastrointestinal symptoms in the context of Autism Spectrum Disorder (ASD). Despite the scientific importance of the subject and the extensive literature review, the manuscript suffers from several fundamental limitations that need to be addressed before it can be approved for publication.

The most significant issue is the length and structure of the text. The manuscript resembles a thesis or monograph rather than a concise and focused scientific publication. The authors attempt to cover an overly broad range of topics, which dilutes the focus on key conclusions and evidence. The paper needs substantial condensation and should center on the most critical aspects, such as the relationship between ARFID and the microbiome or the impact of the gut-brain axis on ASD symptoms.

Another limitation is the lack of a clear methodology and adherence to guidelines for narrative reviews, such as PRIAMS or PRISMA, which diminishes the credibility of the conclusions presented. The authors should adopt PRISMA for the literature selection process and include a flow diagram illustrating the selection process to enhance transparency. This type of visualization would be especially useful given the complexity of the topic.

The article also fails to provide a detailed discussion of the strengths and limitations of the studies analyzed, which is a significant oversight. Including this information would be essential for assessing the reliability of the conclusions. Moreover, the paper relies heavily on secondary data and does not provide sufficient original research findings or evidence.

I recommend the following changes:

  1. Condense the text and focus on the most relevant topics.
  2. Apply PRIAMS or PRISMA guidelines, particularly for literature selection, and include a flow diagram illustrating the selection process.
  3. Explicitly outline the strengths and weaknesses of the studies analyzed.
  4. Expand the discussion of mechanisms and clarify the methods used to analyze the microbiome in the context of ARFID and ASD.
  5. Include future research directions to provide practical applications for the findings.

Without these changes, the article does not meet the standards of a scientific publication and should be subjected to major revisions. Implementing the suggested improvements would make the text more valuable to the scientific community and practitioners.

All the best!

Author Response

Dear Reviewer,

Thank you for your sincerity and for your insightful comments, which demonstrate both depth of knowledge and an understanding of research development.

Indeed, given its length, our work does not strictly adhere to the rigid and precise standards of PRISMA. However, as you pointed out, narrative reviews do not have universally established guidelines.

We believe that this is not merely a limitation but also an advantage. This approach allowed us to comprehensively explore the association between gastrointestinal symptoms and autism while considering the role of the microbiota. While we acknowledge that this method does not yield measurable qualitative outcomes, we are convinced that a systematic approach would not have enabled us to conduct such an analysis.

One of the main reasons for adopting a narrative approach was the need for a multidimensional analysis, which allowed us to interweave evidence from heterogeneous studies to present a complete picture of topics such as microbiota, gastrointestinal issues, and eating disorders in ASD. Given the narrow focus of this subject and the significant methodological variability across studies, a systematic review would not have been feasible (Petticrew, M., Roberts, H. (2006). Systematic reviews in the social sciences: A practical guide.). 

Nonetheless, this does not exempt us from improving the methodological section for greater clarity. With this in mind, I now address your valuable comments:

C1: Condense the text and focus on the most relevant topics.

R1: Thank you for your comment. Each author independently screened the manuscript, and after providing their feedback, we collectively removed all passages deemed unnecessary. However, as we also aimed to address comments from other reviewers, the overall length of the manuscript remains largely unchanged, for which we sincerely apologize. To improve readability, we have extracted brief summaries for each chapter (except Chapters 3 and 4), which should help facilitate comprehension.

C2 and C3: 

Apply PRIAMS or PRISMA guidelines, particularly for literature selection, and include a flow diagram illustrating the selection process.

Explicitly outline the strengths and weaknesses of the studies analyzed.

R2 and R3: Additionally, we have highlighted the significant limitations of our study by incorporating a dedicated discussion section. This replaces the previous summary sections and now explicitly addresses both the limitations of our work and those of the studies included in our review. To ensure clarity and to avoid implying adherence to PRISMA, we have opted not to include a flowchart. However, we have thoroughly revised the methodology section, clarifying the inclusion and exclusion criteria to the best extent possible. Throughout the manuscript, we have consistently emphasized the limitations of the narrative approach.

C4: Expand the discussion of mechanisms and clarify the methods used to analyze the microbiome in the context of ARFID and ASD.

R4: Furthermore, we have expanded the discussion on mechanisms and revised the respective chapters accordingly. Please see the attached revised version.

C5: Include future research directions to provide practical applications for the findings.

R5: In the new summary of the paper, we have included additional information to suggest potential future research directions and have sought to more explicitly highlight the clinical implications of our findings.

We recognize that our work takes a multidirectional approach, which may set it apart. However, we hope that the corrections and refinements made sufficiently address your concerns and enhance the clarity and rigor of our manuscript.

Best regards and thank you for your work,

Stefan Modzelewski and coauthors

Reviewer 3 Report

Comments and Suggestions for Authors

Review of the Manuscript: Unraveling the Connections: Eating Issues, Microbiome, and Gastrointestinal Symptoms in Autism Spectrum Disorder

General Evaluation:

The manuscript addresses an important topic: the connection between Autism Spectrum Disorder (ASD), dietary habits, and gastrointestinal issues, which is relevant for improving health outcomes in ASD individuals. The paper provides a comprehensive review, integrating multiple facets like the microbiome-gut-brain axis, food selectivity, and therapeutic interventions. While the writing is generally clear, some sections are overly dense, making it harder for readers to follow. Repetition of key concepts (e.g., food selectivity and the gut-brain axis) occurs in several sections.The manuscript is well-structured, following a logical flow from the introduction to therapeutic strategies. However, certain sections, such as "Dietary Patterns in Children with ASD," overlap with other areas and could be streamlined. Proper formatting and referencing are consistent, but some figures lack detailed legends, which makes them harder to interpret.

Detailed line-by-line analysis:

# Abstract:

-Line 13. The phrase "Various eating disorders" is vague. Specify examples (e.g., ARFID, food neophobia) for clarity.

-Line 20. The term "vicious cycle" needs elaboration to ensure clarity for readers unfamiliar with the concept.

-Lines 24-25. The dietary interventions (e.g., ketogenic diet) are listed without context. Briefly mention their purpose or benefits.

# Introduction:

-Lines 36-38. The prevalence statistics for ASD need a citation to validate the claim.

-Line 43. "Five times more likely" should be supported by a reference to ensure accuracy.

-Lines 49-50. The term "behavioral rigidity" is introduced but not explained. Briefly define it or provide an example.

# Methods

-Line 73. Mention why formal systematic review methodology was not employed. This strengthens the transparency of the approach.

-Lines 70-74. Specify the inclusion/exclusion criteria for studies to make the methodology more robust.

#Results and Discussion

-Lines 89-95. Redundant information on food group rejection overlaps with later sections. Consider consolidating these findings.

-Lines 173-176. Subtype descriptions for ARFID are informative but could benefit from a summary table for clarity.

-Lines 207-210. The chronic nature of food selectivity is emphasized without suggesting potential interventions. Add a reference to potential solutions.

# Figures and Tables

-Figure 1 (Line 303). The visual is informative but lacks detailed captions explaining the implications of the data.

-Table 1 (Line 797). Include references directly within the table for transparency and ease of cross-referencing.

#Conclusion

- Lines 796-799. The recommendation to "break the cycle" is too broad. Provide specific examples of how targeted interventions can address key stages.

# Suggestions for Improvement

-Abstract. Provide more context for the dietary interventions mentioned and clarify the scope of the review.

-Introduction. Streamline background information to focus on the unique contributions of this review.

- Methods. Include a brief rationale for the databases and search terms used. Explain the choice of narrative review over systematic review in more detail.

- Results and Discussion. Reduce redundancy in the description of ARFID and food selectivity. Add subheadings in the discussion for better readability (e.g., "Impact on Nutritional Deficiencies," "Therapeutic Strategies").

- Figures and Tables. Revise figure legends to include more explanatory details. Add a new table summarizing key findings related to dietary interventions and their outcomes.

- Conclusion. Make the conclusion more action-oriented by specifying practical implications for researchers and clinicians.

Author Response

Dear Reviewer,

Thank you for your time and valuable comments. We are pleased that you find the structure of our work logical. In response to your feedback, we have implemented several corrections to enhance the readability of our review. Where possible, we have shortened the text and included brief summaries at the end of each chapter to facilitate comprehension. We hope these modifications improve the overall clarity and coherence of our work.

Line 13: The phrase "Various eating disorders" is vague. Specify examples (e.g., ARFID, food neophobia) for clarity.
R1: Thank you for this comment. We have applied the necessary corrections.

Line 20: The term "vicious cycle" needs elaboration to ensure clarity for readers unfamiliar with the concept.
R2: The term has been clarified.

Lines 24-25: The dietary interventions (e.g., ketogenic diet) are listed without context. Briefly mention their purpose or benefits.
R3: We have added background information, which should better justify the inclusion of dietary interventions.

Lines 36-38: The prevalence statistics for ASD need a citation to validate the claim.
R4: Thank you for this comment. We have added the necessary citations.

Line 43: "Five times more likely" should be supported by a reference to ensure accuracy.
R5: We have added the necessary references.

Lines 49-50: The term "behavioral rigidity" is introduced but not explained. Briefly define it or provide an example.
R6: We have clarified this concept.

Line 73: Mention why formal systematic review methodology was not employed. This strengthens the transparency of the approach.
R7: Thank you for this comment. We have emphasized the reason for this approach and rewritten the methodology section.

Lines 70-74: Specify the inclusion/exclusion criteria for studies to make the methodology more robust.
R8: To the extent possible for a narrative review, we have specified the inclusion and exclusion criteria.

Lines 89-95: Redundant information on food group rejection overlaps with later sections. Consider consolidating these findings.
R9: We have removed redundant information; thank you for this comment.

Lines 173-176: Subtype descriptions for ARFID are informative but could benefit from a summary table for clarity.
R10: We have created an appropriate table, which is included in the manuscript. Please see the attached file.

Lines 207-210: The chronic nature of food selectivity is emphasized without suggesting potential interventions. Add a reference to potential solutions.
R11: We have added relevant references to potential solutions.

Figure 1 (Line 303): The visual is informative but lacks detailed captions explaining the implications of the data.
R12: We have improved the caption.

Table 1 (Line 797): Include references directly within the table for transparency and ease of cross-referencing.
R13: We have added references to the tables; thank you for these comments.

Lines 796-799: The recommendation to "break the cycle" is too broad. Provide specific examples of how targeted interventions can address key stages.
R14: We have significantly expanded the conclusion to include relevant implications.

  • Abstract: Provide more context for the dietary interventions mentioned and clarify the scope of the review.
  • Introduction: Streamline background information to focus on the unique contributions of this review.
  • Methods: Include a brief rationale for the databases and search terms used. Explain the choice of narrative review over systematic review in more detail.
  • Results and Discussion: Reduce redundancy in the description of ARFID and food selectivity. Add subheadings in the discussion for better readability (e.g., "Impact on Nutritional Deficiencies," "Therapeutic Strategies").
  • Figures and Tables: Revise figure legends to include more explanatory details. Add a new table summarizing key findings related to dietary interventions and their outcomes.
  • Conclusion: Make the conclusion more action-oriented by specifying practical implications for researchers and clinicians.

R15: Thank you for summarizing the feedback. We have made an effort to address all points. Please see the attached document.

Thank you for your work,

Kind regards,
Stefan Modzelewski and coauthors.

Round 2

Reviewer 2 Report

Comments and Suggestions for Authors

Dear Authors,

The revised version of the article has made significant changes that improve its quality, such as enrichment with diagrams and a more detailed discussion of the microbiome in the context of ARFID and ASD. However, the text still needs revisions to reach a publishable level. The following are recommendations:

1. the authors argue that in narrative reviews the full use of PRISMA is not necessary. I agree that such reviews are more casual in nature, but this does not relieve the transparency of the literature selection process. To increase the credibility of the paper, a literature selection flow chart should be developed (even in a simplified form) that clearly shows the number of studies searched, selected and excluded, and the reasons for exclusion. This approach is universal good practice and does not limit the narrative nature of the review. The criteria for including and excluding studies should be described in more detail, preferably in a table, to allow a better understanding of the selection process.

2 The article still lacks a clear discussion of the strengths and weaknesses of the studies analyzed. Without such an analysis, the conclusions may seem one-sided. Recommendations:

a. Include a section assessing the quality of the evidence, addressing potential methodological flaws, limitations in the heterogeneity of the studies, and possible sources of bias.

b. Provide statistical data, such as confidence intervals or p-values, where possible, to increase the credibility of conclusions.

3. Despite the changes made, the text is still too long and feels more like a monograph than a scientific article. Recommended changes:

a. Shorten the introduction and general sections on ASD, which are well known, and focus on more niche aspects, such as the effects of ARFID on the microbiome and mechanisms of the microbiome-gut-brain axis.

b. Remove repetitive content, especially in sections describing microbial mechanisms.

4. Although the introduction of a more detailed discussion of SCFAs and the microbiota is valuable, the conclusions are still not fully integrated with the clinical implications. Recommendations:

a. Clearly link biological mechanisms to ASD symptoms and therapeutic suggestions.

b. Take a more critical look at the effectiveness of the interventions discussed, such as ketogenic diets and probiotics. Their long-term effects, limitations and risks should be considered.

5. The article should include more specific recommendations for future research, such as:

What types of intervention studies are needed?

What groups of patients should be analyzed?

What research methods could help to better understand the impact of microbiota on ASD?

6 The text should be more concise and devoid of overly technical language in sections aimed at a broad audience.

Although the changes have improved the article's quality, key elements are missing to consider it ready for publication. Until the guidelines for transparency in the literature selection process and critical appraisal of evidence are addressed, the article should be regarded as in need of further revisions. Meeting the above recommendations will significantly increase its value to the scientific community and clinical practitioners.

Author Response

Dear Reviewer,

Thank you for your insightful comments and suggestions. We greatly appreciate your feedback, which has been invaluable in improving the quality of our manuscript. Below, we address your comments in detail:

R1:
We appreciate this observation and have worked to enhance our methodology accordingly. A flowchart has been added to the manuscript to clearly present the study selection process.

R2:
To ensure a more objective presentation of the data and to avoid overemphasizing the role of the microbiota in the disorder, we have included Table S1 as a supplementary material. This table summarizes the key findings and limitations of the included studies, helping to contextualize the current evidence.

R3:
Following your suggestion, each author reviewed the manuscript independently. Subsequently, one author undertook the task of streamlining the text after a thorough team discussion. We believe the revised version is now more concise and precise.

R4:
We have removed information that lacked sufficient correlation and rewritten the summary to better align with the discussion section. Wherever applicable, we enriched the content with mechanistic insights. Additionally, we emphasized the limitations of therapeutic approaches targeting the microbiota to avoid inadvertently promoting alternative treatments over established evidence-based therapies.

R5:
In the conclusion, we have proposed directions for future research, outlining specific areas that warrant further investigation.

R6:
To improve the logical flow and clarity of the manuscript, we have reorganized its structure. One chapter has been repositioned to create a more coherent cause-and-effect sequence, with background information now preceding the related findings.

We are confident that the revisions address the concerns raised and result in a stronger manuscript. Thank you again for your thoughtful comments and for the opportunity to revise our work.

Sincerely,

Stefan Modzelewski and coauthors
